# Barriers and facilitators to antibiotic stewardship in Nigeria's private healthcare sector: A qualitative interview study with national health and regulatory interest holders

Aaya Mahdi[1], Sarah Pascale Ngassa Detchaptche[1], Joel Shyam Klinton[2], Nnakelu Eriobu[3], Blessing Uche[3], Nawal Maredia[1], Charity Oga-Omenka[2,3], Giorgia Sulis[1,3,4]*

**1** Faculty of Medicine, School of Epidemiology and Public Health, University of Ottawa, Ottawa, Canada, **2** Faculty of Health, School of Public Health Sciences, University of Waterloo, Waterloo, Canada, **3** International Research Centre of Excellence (IRCE), Institute of Human Virology Nigeria (IHVN), Abuja, Nigeria, **4** Methodological and Implementation Research Program, Ottawa Hospital Research Institute, Ottawa, Canada

* gsulis@uottawa.ca

## Abstract

Antimicrobial resistance (AMR) poses a growing global health concern, particularly in low- and middle-income countries (LMICs) such as Nigeria, where stewardship efforts have largely overlooked the private healthcare sector. Given that most Nigerians seek care from private providers, understanding the barriers and opportunities for implementing antibiotic stewardship programs in this setting is critical. We conducted a qualitative study consisting of semi-structured interviews with 14 interest holders from national health and regulatory organizations in Nigeria. Participants were purposively selected for their expertise and leadership roles related to antibiotic use and regulation in Nigeria, ensuring representation from professional associations, federal agencies, and technical bodies. Interviews were conducted virtually between August and December 2024, transcribed verbatim, and thematically analyzed using Braun and Clarke's six-step framework. Codes were grouped into four thematic domains reflecting the roles of key actors (private sector, practitioners, government and health systems, and patients), and mapped to the Social-Ecological Model (SEM) to reflect individual, organizational, and policy-level influences. Participants identified multiple barriers to stewardship implementation in the private sector, including inconsistent prescribing practices, limited access to diagnostics, weak regulatory enforcement, and knowledge gaps among practitioners and patients. System-level constraints such as workforce shortages and fragmented AMR surveillance infrastructure were also cited. However, participants did highlight several facilitators, including licensing-linked training requirements, institutional partnerships, and openness within the private sector to adopt stewardship initiatives. Recommended strategies included

**Data availability statement:** All data and meta-data underlying reported findings are provided within the paper and its Supporting Information files. Specifically, S3 File contains excerpts from all interviews regarding all themes and subthemes described in the paper, and select quotes are included in the main article. However, given the nature and number of participants, full interview transcripts are not made publicly available to protect confidentiality.

**Funding:** This project was supported by a seed grant from the University of Ottawa's Research Program Development Fund awarded to GS. The work was also partly supported by the Tier 2 Canada Research Chair in Communicable Disease Epidemiology held by GS. The funders had no role in study design, data collection and analysis, decision to publish, or preparation of the manuscript.

**Competing interests:** I have read the journal's policy and the authors of this manuscript have the following competing interests: GS serves as Academic Editor for PLOS Global Public Health. All other authors have declared that no competing interests exist.

standardized training programs, development of prescribing guidelines and protocols, performance-based incentives, and community-based public awareness campaigns. This study underscores the urgent need for targeted, context-sensitive stewardship strategies tailored to Nigeria's private healthcare sector. Engagement with interest holders revealed both systemic challenges and actionable opportunities to strengthen antibiotic stewardship and support broader AMR control efforts.

## Introduction

Antimicrobial resistance (AMR) is a growing global health crisis [1], with drug-resistant bacterial infections responsible for an estimated 5 million deaths annually and projections suggesting this number could rise to 10 million by 2050 if no coordinated global action is taken [2]. The burden of AMR is disproportionately higher in low- and middle-income countries (LMICs), where up to 90% of global AMR-related deaths occur [3]. In these settings, weak surveillance systems, limited healthcare funding, and restricted access to effective antibiotics undermine efforts to regulate antibiotic use and control resistance [4].

Nigeria, the most populous country in Africa [5], faces a particularly urgent AMR challenge. Drug-resistant bacterial infections account for more than 27 deaths per 100,000 people annually in West Africa, where Nigeria is located [2]. The country's decentralized health system includes a large private sector that operates independently from government-regulated facilities [6]. Private healthcare providers in Nigeria deliver 60% of health services yet operate only 30% of healthcare facilities [7–8]. Public healthcare facilities suffer from qualified staff shortages, which drives patients to seek care from private providers as a first point of access [6]. Despite this, national AMR strategies (including stewardship efforts) have largely focused on public hospitals, leaving the private sector underrepresented and poorly integrated [9].

Several systemic and behavioral factors contribute to AMR in Nigeria. Antibiotics are frequently dispensed without prescription, and financial incentives may drive practitioners to prioritize sales over clinical need [4]. Additionally, misconceptions about antibiotic use and effects are widespread, leading patients to demand unnecessary prescriptions or self-medicate with leftover drugs, due to a desire for quick symptom relief or prior use of antibiotics for similar ailments [10]. These issues are compounded by lack of awareness, gaps in AMR training among providers, and limited access to diagnostics [10].

The primary goal of antibiotic stewardship programs is to promote proper use of antibiotics and slow the emergence and spread of resistance [11]. These initiatives support providers in tailoring treatments to individual patient needs and ensuring that prescribing decisions align with best practices [9]. Although many antimicrobial stewardship programs have demonstrated success globally and in various LMICs – including Ghana, India, and South Africa [12–14] – these initiatives have largely focused on the public health sector. In Nigeria, most interventions remain confined to tertiary public hospitals. Community-based private practitioners who are often the first point of contact for healthcare-seeking individuals, have been largely excluded from these efforts [9]. This gap represents a missed opportunity to address AMR at scale.

To design effective, context-sensitive stewardship interventions, it is essential to understand the perspectives of national interest holders involved in AMR policy, regulation, and implementation. A qualitative approach is well-suited to capturing the complex social, institutional, and systemic factors that shape antibiotic use and stewardship feasibility in Nigeria's private sector. This study aimed to explore the barriers, facilitators, and opportunities for implementing antibiotic stewardship interventions in Nigeria's private healthcare sector, based on interviews with key national interest holders.

## Methods

This qualitative study employed semi-structured interviews to gather in-depth perspectives from professionals involved in health policy, regulation, and service delivery. Given the complexity of stewardship design and implementation, this approach allowed for the collection of rich, context-specific insights into the systemic, institutional, and behavioral dynamics shaping stewardship acceptability and feasibility in the private sector.

For reporting, we followed the Consolidated Criteria for Reporting Qualitative Research (COREQ) framework, and the completed checklist is available in S1 File.

### Study participants

Fifteen individuals were purposively selected and invited to participate to ensure representation from key national and regional institutions including the Federal Ministry of Health and Social Welfare, the Medical and Dental Council of Nigeria, the Nigeria Centre for Disease Control and Prevention, and the World Health Organization's Country Office (Table 1). Participants held leadership or technical roles in their respective organizations, thus providing insights on challenges and opportunities concerning private-sector antibiotic prescribing and stewardship implementation. The research team at the University of Ottawa and the Institute of Human Virology Nigeria (IHVN) contacted potential participants via email and by phone to provide them with study details and share the informed consent form.

### Data collection

Interviews were conducted remotely in English between 13 August 2024 and 13 December 2024, using Zoom (2025 Version 6.3.11) or Microsoft Teams (2025 Version 25031). A trained qualitative research consultant initially led the interviews alongside a research assistant and a student researcher, who later conducted interviews independently after completing training.

**Table 1. Organizations represented by interviewed participants.**

| Organization Name |
| --- |
| Association of Nigerian Private Medical Practitioners (ANPMP) – Abuja |
| Association of Nigerian Private Medical Practitioners (ANPMP) – Federal Capital Territory |
| Association of Nigerian Private Medical Practitioners (ANPMP) – Lagos |
| Federal Ministry of Health and Social Welfare (FMOHSW) |
| Fleming Fund, Nigeria |
| International Foundation Against Infectious Diseases in Nigeria (IFAIN) |
| Medical and Dental Council of Nigeria (MDCN) |
| National Association of Government and General Medical and Dental Practitioners (NAGGMDP) |
| National Association of Resident Doctors (NARD) |
| National Postgraduate Medical College of Nigeria (NPMCN) |
| National Primary Health Care Development Agency (NPHCDA) |
| Nigeria Centre for Disease Control and Prevention (NCDC) |
| Nigerian Medical Association (NMA) – Federal Capital Territory |
| World Health Organization (WHO) |

The interview guide, developed a priori, covered topics such as the participant's knowledge of and perceptions about AMR, the role of the private healthcare sector in AMR response, barriers and facilitators to implementing stewardship interventions, as well as opportunities and potential solutions for improving prescribing practices and provider engagement (see S2 File). Interviews lasted 30–60 minutes, were audio-recorded with participant consent, and transcribed verbatim using Zoom and Microsoft Teams auto-transcription. Transcripts were manually checked for accuracy and anonymized prior to analysis.

## Data analysis

Interview data were processed in NVivo (Version 14, 2023) and analyzed thematically in accordance with Braun and Clarke's six-step methodology [15]. The student researcher performed initial coding, generating descriptive labels to represent key ideas expressed in the interview transcripts. These codes were refined iteratively alongside a second research team member who independently reviewed the coded transcripts, discussing any discrepancies and resolving them through consensus. Using Microsoft Excel (MS Excel 2024), descriptive codes were then grouped into broader themes representing key concepts across four domains: (1) private sector, (2) healthcare practitioners, (3) patients, and (4) government and health systems. Each domain included four subthemes: (1) knowledge and perceptions, (2) barriers, (3) opportunities, and (4) strategies. To organize and interpret our findings, we applied a deductive framework based on three relevant levels of the Social-Ecological Model (SEM) [16], which provides a systematic way to map dynamic interactions and interrelations of multilevel factors that influence health behaviors. Specifically, we considered the following SEM levels: individual and interpersonal (practitioners and patients), organizational (private sector), and system/policy (government and health systems). This combined inductive-deductive approach ensured that our analysis remained grounded in participant narratives while aligned with a well-established conceptual model.

While thematic saturation was considered, no formal assessment was conducted. Interview codes categorized by theme and subtheme are presented in S3 File. It should be noted that the interviewers were trained in qualitative methods and worked collaboratively across institutions in Canada and Nigeria. The team engaged in regular reflexive discussions to minimize interpretive bias and contextualize findings within the Nigerian healthcare landscape.

## Ethical considerations

Ethical approval was obtained from the University of Ottawa Research Ethics Board (approval no. H-06-24-10154), the Lagos State Health Research Ethics Committee (approval no. LSHREC/2024/00012), and the Federal Capital Territory Abuja State Health Research Ethics Committee (approval no. FHREC/2024/01/138/13–06024). All participants provided informed consent through an emailed form and verbally confirmed their willingness to participate before the start of the interview. All interviews were conducted in private settings, and data were securely stored on a password-protected University of Ottawa OneDrive folder accessible only to authorized team members.

## Results

A total of 14 interviews were completed with representatives from key organizations in Nigeria. All participants had a medical background and held leadership roles within their respective organizations. Of the 14 participants, 4 were professional association representatives, 6 were senior officers (e.g., secretary or technical officer), and 4 were executives (e.g., manager or director). Fig 1 presents a word cloud generated from the interview transcripts.

## Summary of themes

Our analysis generated four overarching themes reflecting the major groups influencing antibiotic use and stewardship in Nigeria: (1) private sector, (2) healthcare practitioners, (3) patients, and (4) government and health systems. Each theme incorporates subthemes capturing participants' knowledge and perceptions, barriers, opportunities, and strategies related to AMR and stewardship implementation (Table 2).

**Fig 1. Word cloud generated in NVivo from interview transcripts.** This word cloud was generated from the full set of interview transcripts using NVivo (Version 14). It displays all words that appeared at least 50 times across the interviews, including stemmed words (for instance "prescribe", "prescribing", "prescription", were grouped as a single root). The key terms "private sector", "stewardship", "antibiotics", and "Nigeria" appear frequently in the participants' responses.

## Individual & Interpersonal Levels

### Role of individual practitioners

**Barriers to effective stewardship.** Most participants agreed that clinical guidelines are inconsistently followed in the private sector, with two participants noting the use of casual or verbal prescriptions. It was also highlighted how some healthcare workers, including non-clinicians, prescribe based on patient affordability rather than proper protocols. Some participants described trial-and-error prescribing approaches as common, likely being a major contributor to AMR. Notably, one participant emphasized that changing prescribing behaviors remains a major challenge for stewardship as many are resistant to change.

*"Because the private sector is a mix of both those with clinicians and others who are health workers—not laboratory scientists or community health extension workers. Many of them also have small practices on the side, and they provide drugs based on what the patient can afford, without necessarily considering whether the patient will receive adequate dosing."* (**Participant #4 – Executive**)

**Table 2. Summary of emergent themes and subthemes from qualitative interviews with representatives of key organizations involved in Nigeria's health system, organized by Social-Ecological Model (SEM) level.**

| SEM Level | Theme | Subtheme | Description | Example of quote |
|---|---|---|---|---|
| Individual and Interpersonal | Role of individual practitioners | Barriers to effective stewardship | Participants reported inconsistent adherence to clinical guidelines, profit-driven prescribing, and low awareness as key barriers to stewardship, with limited training access and resistance to behavior change further hindering appropriate access. | *"Well, it's really looking at maybe other challenges as well. A level of awareness, and it's gonna be a challenge because now even for the public facility, like I mentioned, the people that we are training—it's not all the people in the space that we are training. We're just training a few people per facility."* **(Participant #6 – Executive))** |
| | | Opportunities for AMR-related activities in the private sector | Current diagnostic testing and licensing-linked training requirements were key opportunities identified to strengthen stewardship practices among private practitioners. | *"What I do before I place a patient on any antibiotics is normally send them for a culture—that is, when I suspect that the patient has septicemia or an infection. I usually send them for a culture, and while awaiting the culture report, I normally place them on a broad-spectrum antibiotic"* **(Participant #1 – Senior Officer)** |
| | | Strategies to enhance private practitioners' engagement in AMR response | Participants emphasized the importance of expanding standardized training, strengthening patient-provider communication, increasing in-facility awareness efforts, and implementing SOPs as key actions to improve private practitioner involvement in AMR stewardship. | *"So, this SOP will guide on the right things to be done. Healthcare providers in primary healthcare institutions, such as private institutions, can work within these SOPs to guide them. And then, this will help in further reducing, antimicrobial resistance and its associated complications"* **(Participant #11 – Senior Officer)** |
| | Role of patients | Patient-related barriers | Patients' financial constraints, low awareness, and harmful treatment practices (sharing antibiotics, purchasing incomplete doses) were consistently cited as key drivers of inappropriate antibiotic use. | *"The challenge again is that many of them—finance is an issue. Because when they come to the hospital, they have to take a card to consult with a doctor. They first of all prefer to go to a patent medicine store or go to a pharmacy where they will not take a card or consult anybody. They just buy the drug, so they prefer to go there. It's only when they don't have a solution to the problem that they will now consult with a doctor."* **(Participant #1 – Senior Officer)** |
| | | Suggested strategies | Participants highlight the need for broad public education through mass media, community engagement, and the involvement of healthcare professionals to promote responsible antibiotic use and reduce misinformation. | *"Except for government intervention. The government can use social media, radio, or television for awareness creation in religious houses, so the population will be well-informed."* **(Participant #1 – Senior Officer)** |
| Organizational | Role of private sector | The private sector as a key player in Nigeria's health sector | The private sector was consistently identified as the primary access point for healthcare in Nigeria, with participants emphasizing its central role in antibiotic use and its potential to drive stewardship efforts. | *"…I think the role is very big. In any country, the private sector has a significant role. In Nigeria, the private sector is crucial because most people prefer to go there… Since most individuals go to the private sector, that's where the majority of antimicrobials are used. The private sector's role is critical in minimizing the impact of antimicrobial resistance."* **(Participant #11 – Senior Officer)** |
| | | Barriers to private sector engagement in AMR response | Participants noted that stewardship activities remain largely absent in the private sector, with limited representation and inclusion in coordinated AMR response efforts. | *"…in general, there is some progress that we can see in the country regarding antimicrobial stewardship. But please note that all I'm making reference to is just with the government hospitals. The private sector still has a lot that needs to happen"* **(Participant #14 – Executive)** |
| | | Facilitators of AMR stewardship in the private sector | Participants described the private sector as receptive to stewardship initiatives, with institutional training and established partnerships with regulatory bodies identified as key enablers of AMR engagement. | *"In terms of capacity building in the private sector for antimicrobial stewardship, this happens in local health facilities where they have their grand rounds, seminars, conferences, and peer review meetings. So this may happen in individual health institutions."* **(Participant #3 – Executive)** |

*(Continued)*

**Table 2.** (Continued)

| SEM Level | Theme | Subtheme | Description | Example of quote |
|---|---|---|---|---|
| System/Policy | Role of government and health systems | Perceptions on the burden of AMR | AMR was described as a serious issue with severe global health and economic consequences in Nigeria, reflecting its national burden. | *"In Nigeria, the prevalence is quite high. You know, research has shown that in this part of the world—for example, a study in 2019 found that over, I think, over yeah, over 65,000—64,000,—deaths were directly related to antimicrobial resistance."* (**Participant #11 – Senior Officer**) |
| | | System-level barriers | Participants described barriers to AMR control, including diagnostic and workforce gaps, poor enforcement of regulations, weak surveillance systems, and unregulated antibiotic access. These factors collectively hinder stewardship implementation and drive resistance. | *"So, one of the challenges actually? The doctor-patient ratio in Nigeria is very poor. Realistically, a lot of people that need to see a doctor may end up not seeing one. This leads to a situation where somebody who is not a doctor, but claims to have some knowledge of patient care, ends up prescribing medication"* (**Participant #10 – Senior Officer**) |
| | | Opportunities | There is growing recognition of national and global support for AMR efforts in Nigeria, with emerging opportunities linked to improved surveillance, digital data sharing platforms, and structured stewardship initiatives supported by collaborative efforts. | *"For Nigeria, we have been doing AMR surveillance since 2017, even though it is not very optimal, and our data is not as representative as we would expect. But then we have started something."* (**Participant #2 – Senior Officer**) |

The table highlights identified barriers, opportunities, and strategies for antibiotic stewardship across the private sector, healthcare practitioners, government and health systems, as well as patients.

Abbreviations: AMR, Antimicrobial resistance; SEM, Social-Ecological Model.

*"Because when you're talking about stewardship, it entails coordination to promote optimal use of all these drugs, you understand, drug choices. So it's like a trial and error, really, in that space. So once the client comes, I give the person amoxicillin, and if this person still comes back, I feel I need to give a stronger antibiotic. It's all about the patient being able to afford these payments. The more they can pay, the more the... the better they get."* (**Participant #6 – Executive**)

*"Even if the evidence is there that, yes, there's a need to improve the way antibiotics are used, and there's support that can be provided to the facility, there's still the real challenge of how we'll actually get people to change their behaviors over time. There needs to be a deliberate effort to provide mentorship to private facilities."* (**Participant #14 – Executive**)

Limited access to stewardship training and low awareness among practitioners also emerged as relevant barriers to appropriate antibiotic use from our participants' perspective. Two participants noted that only a few individuals per facility receive stewardship training, even in the public sector, explaining that clinicians often skip training to avoid losing income from clinical duties. The same participants added that many prescribers lack the general knowledge required to follow evidence-based practices.

*"Well, it's really looking at maybe other challenges as well. A level of awareness, and it's gonna be a challenge because now even for the public facility, like I mentioned, the people that we are training—it's not all the people in the space that we are training. We're just training a few people per facility."*

(**Participant #6 – Executive**)

*"In terms of barriers for the prescribers and then barriers to the health sector in general, I mean... Well, one of the barriers might be the aspect of not knowing."* (**Participant #8 – Senior Officer**)

**Opportunities for AMR-related activities in the private sector.** Diagnostic testing was seen as a critical step to guide rational antibiotic use. Participants emphasized the importance of collecting samples for culture testing before initiating antibiotic treatment. As one participant explained, broad-spectrum antibiotics are often used as a temporary measure while awaiting culture results.

*"And… encouragement of people to run tests before they start taking antibiotics. And patients should be encouraged to complete the antibiotics for the duration they are supposed to take the drugs."* (**Participant #1 – Senior Officer**)

*"What I do before I place a patient on any antibiotics is normally send them for a culture—that is, when I suspect that the patient has septicemia or an infection. I usually send them for a culture, and while awaiting the culture report, I normally place them on a broad-spectrum antibiotic."* (**Participant #1 – Senior Officer**)

In addition, participants noted that medical licensing renewal requirements offer a vehicle for scaling up provider training. Relatedly, one participant noted that national and international modules, such as those from WHO, exist but have not been fully scaled. As well, though public facilities remain the primary focus, participants noted the private sector was included in previous national training efforts.

*"There are routine trainings to update ourselves. For instance, before you renew your medical practicing license every year, you must have attended some trainings. There's a minimum score you must have before they can issue you a practicing license. This could be done online."* (**Participant #1 – Senior Officer**)

**Strategies to enhance private practitioners' engagement in AMR response.** Participants frequently emphasized that ongoing education programs and accessible training materials are essential to upskill healthcare providers. They highlighted the need for standardized training programs to better involve the private sector. Another participant added that training efforts must account for staff disparities and access between rural and urban regions.

*"… there might be a need for trainings, particularly at the community level, so they can become accustomed to the standard way. At the tertiary institutions, some of these practices are being implemented, but if you go lower down to the secondary or primary levels, and especially in the private sector, you'll find that all of those measures are not in place."* (**Participant #4 – Professional Association Representative**)

Four participants indicated that prescribers play a central role in encouraging patients to complete the full prescribed course of antibiotics and in discouraging indiscriminate use. One participant emphasized that private sector clinicians must actively explain and reinforce responsible antibiotic use among patients. Other participants noted that practitioner-tailored awareness campaigns at the healthcare facility level, such as posters or internal seminars, could help support adherence to clinical guidelines. Additionally, the adoption of standard operating procedures (SOPs) was viewed as essential to guide prescriber behavior and promote evidence-based care.

*"But we, the private practitioners, we hold them a responsibility to explain to them. But, you know, we don't force them. Like I explained earlier, they have a right to take a decision, but I will let them know the points or components of whatever decision they are taking."* (**Participant #1 – Senior Officer**)

*"For administering antibiotics, they will carry out preliminary investigations and all that. Then they start with an empirical antibiotic prospect, and once the culture and sensitivity results are out, they can now switch antibiotics."* (**Participant #9 – Professional Association Representative**)

*"So, I will embark on that [advocacy] amongst others. One of the other things I will do is to put in place—I think I mentioned this earlier—standard operating procedures."* (**Participant #11 – Senior Officer**)

*"So, this SOP will guide on the right things to be done. Healthcare providers in primary healthcare institutions, such as private institutions, can work within these SOPs to guide them. And then, this will help in further reducing, antimicrobial resistance and its associated complications."* (**Participant #11 – Senior Officer**)

### Role of patients

**Patient-related barriers.** Patients' self-medication, incomplete antibiotic courses, and reliance on unregulated drug vendors were identified as major contributors to AMR. Financial hardship and lack of awareness were seen as driving these behaviors. Of note, nine participants reported that non-adherence to prescriptions is common, with many patients failing to complete their antibiotic treatments or sharing leftover antibiotics, which reduces effectiveness and contributes to resistance. Some participants noted that patients often self-prescribe or purchase antibiotics from unregulated vendors or without a prescription. One participant highlighted that patient impatience and pressure on providers can lead to inappropriate prescribing before proper diagnostic test results are obtained. Three participants noted that religious and cultural beliefs may also influence treatment decisions, with some patients rejecting antibiotics.

*"So, if another person is having a similar issue around him or her, they prefer to give that leftover antibiotics to that individual. So these are part of the challenges that we have."* (**Participant #1 – Senior Officer**)

*"Some of them are not compliant. If a patient has an issue and you tell him or her to take an antibiotic either twice daily or three times daily, once the patient gets better, they don't complete the antibiotics."* (**Participant #1 – Senior Officer**)

*"Maybe they are experiencing diarrheal disease, and they go in and say, "Give me flagyl," you know, metronidazole. They are given 2-3-4 tablets, take them once, feel better, and stop—not realizing that this behavior contributes to the development of antimicrobial resistance over time."* (**Participant #4 – Professional Association Representative**)

Participants revealed that patients often avoid doctor visits and self-prescribe antibiotics due to the inability to afford full treatments. The majority of participants agreed that financial limitations lead patients to purchase partial antibiotic doses or opt for cheaper alternatives, including counterfeit drugs.

*"The challenge again is that many of them—finance is an issue. Because when they come to the hospital, they have to take a card to consult with a doctor. They first of all prefer to go to a patent medicine store or go to a pharmacy where they will not take a card or consult anybody. They just buy the drug, so they prefer to go there. It's only when they don't have a solution to the problem that they will now consult with a doctor."* (**Participant #1 – Senior Officer**)

*"This is all-encompassing. Unfortunately, the country has been experiencing lots of economic downturns and economic challenges, which are directly affecting the populace. And that is the honest truth. So, poverty is a major challenge that has prevented patients from, you know, doing the right thing. Ignorance is another. I mean, poverty and ignorance go hand in hand."* (**Participant #11 – Senior Officer**)

They also reported that many patients have a limited understanding of when antibiotics are necessary and of the risks associated with misuse. Some added that common misunderstandings about antibiotics often lead to inappropriate use.

*"But the awareness about—or the thought, the awareness about—the antimicrobial usage is not high in Nigeria."* (**Participant #1 – Senior Officer**)

*"Also, many of these patients are not even well-enlightened. They are not well-educated about antimicrobial resistance and its adverse complications or adverse effects."* (**Participant #11 – Senior Officer**)

**Suggested strategies.** Participants emphasized the potential of mass media, social media, and community-level outreach to improve public awareness. They also highlighted the important role of healthcare workers in building awareness and promoting responsible antibiotic use.

"Except for government intervention. The government can use social media, radio, or television for awareness creation in religious houses, so the population will be well-informed." (**Participant #1 – Senior Officer**)

"So, they will also, you know, being professionals, you know, they will be, you know, they will be enlightened in this area. And that will also help in cascading the message down, you know, to their patients, to those in the community." (**Participant #8 – Senior Officer**)

## Organizational Level

### Role of the private sector

**Barriers to private sector engagement in AMR response.** Participants consistently identified the private sector as the primary point of care for most Nigerians and emphasized its central role in antibiotic use and stewardship initiatives.

*"The role of the private sector cannot be overemphasized. Because in Nigeria, the private sector has a very big number of clients because more than 70–75 or 80% of Nigeria's healthcare delivery is out of pocket."* (**Participant #2 – Senior Officer**)

*"…I think the role is very big. In any country, the private sector has a significant role. In Nigeria, the private sector is crucial because most people prefer to go there… Since most individuals go to the private sector, that's where the majority of antimicrobials are used. The private sector's role is critical in minimizing the impact of antimicrobial resistance."* (**Participant #11 – Senior Officer**)

However, three participants acknowledged that stewardship activities are largely absent in the private sector, and one of them noted that private actors are underrepresented in national AMR technical working groups.

*"...in general, there is some progress that we can see in the country regarding antimicrobial stewardship. But please note that all I'm making reference to is just with the government hospitals. The private sector still has a lot that needs to happen."* (**Participant #14 – Executive**)

*"There is a nice technical working group at the national level for stewardship, but I don't see even there in the private sector."* (**Participant #5 – Senior Officer**)

**Facilitators of AMR stewardship in the private sector.** Four participants described the private sector as receptive and willing to adapt to new initiatives, with two participants noting the existence of tailored institutional meetings and internal training activities supporting AMR efforts. Also, a few participants challenged the view that private sector awareness is low, stating that major institutions are already engaged in some awareness activities.

*"...no, I've not come across [challenges]... yes I've seen fair acceptability, I don't encounter any resistance."* (**Participant #1 – Senior Officer**)

*"[The private sector] are amenable. In other words, they are happy to... they are open to receive any new message that comes, especially with regards to improving the healthcare provision."* (**Participant #8 – Senior Officer**)

*"In terms of capacity building in the private sector for antimicrobial stewardship, this happens in local health facilities where they have their grand rounds, seminars, conferences, and peer review meetings. So this may happen in individual health institutions."* (**Participant #3 – Executive**)

Furthermore, ten participants highlighted the role of regulatory bodies in supporting national stewardship efforts, and eight of them referenced joint efforts with regulatory bodies and how these partnerships build capacity for stronger AMR initiatives. Interestingly, one participant mentioned the COVID-19 response effort as evidence of the private sector's ability to collaborate well. National committees were also mentioned by four participants as being active in intervention efforts.

*"WHO, of course, is one of the key technical agencies. Over the last two years, we have done a lot in Nigeria, especially in the development of the national action plan. Through this, WHO did support not only technically but also financially."* (**Participant #5 – Senior Officer**)

*"I'm in a meeting with the NCDC, and they are actually the custodians when it comes to antimicrobial stewardship, bringing in other partners to really coordinate the effort of promoting the optimal use of these antimicrobial agents."* (**Participant #6 – Executive**)

*"When we tried to implement wide-scale testing for COVID-19, the government realized that it wasn't something that could be done alone by the public sector. So, they had to engage with the private sector, and that was one model that I see has worked very well."* (**Participant #14 – Executive**)

## System/Policy Level

### Role of government and health systems

**Perceptions on the burden of AMR.** All participants agreed that AMR is widespread and contributes to severe health complications and increasing healthcare costs. AMR was consistently recognized as a growing global health security threat, with Nigeria identified as being heavily affected.

*"Well, let me start globally. Antimicrobial resistance response and activities are quite new, but because of the growing global momentum to address antimicrobial resistance, a lot of organizations, a lot of government agencies, and development partners are interested in AMR to ensure that it is contained before it explodes as a pandemic."* (**Participant #2 – Senior Officer**)

*"In Nigeria, the prevalence is quite high. You know, research has shown that in this part of the world—for example, a study in 2019 found that over, I think, over yeah, over 65,000—64,000—deaths were directly related to antimicrobial resistance."* (**Participant #11 – Senior Officer**)

**System-level barriers.** Participants identified significant challenges including limited diagnostic infrastructure, workforce shortages, weak policy enforcement, unregulated drug distribution, and surveillance gaps. These systemic issues were seen as key drivers of antimicrobial misuse.

Specifically, participants reported that antibiotic availability is inconsistent, with some drugs being either inaccessible or overused. Some noted that limited laboratory and diagnostic capacity hinders effective AMR control.

*"… gaps in laboratory capacity, as well as to identify the source of the infection. You know, in the laboratories, there are some gaps that have been documented as to identifying sources. […] I mentioned first access to effective antimicrobial agents. We have gaps in funding to carry out planned solutions to wait."* (**Participant #11 – Senior Officer**)

*"Apart from regulation […] I think the access to antibiotics is a big issue. As you know, not all antibiotics are there, and those that are there also need to have rational use. So, there are both ways: access and excess."* (**Participant #5 – Senior Officer**)

Nearly all participants mentioned the widespread shortage of trained healthcare professionals as an important barrier. One participant explained that low incentives and poor working conditions hinder staff retention, while three others described high turnover of health workers seeking better opportunities abroad. In the private sector, dual-role staffing challenges were noted, with some highlighting uneven workforce distribution.

*"So, one of the challenges actually? The doctor-patient ratio in Nigeria is very poor. Realistically, a lot of people that need to see a doctor may end up not seeing one. This leads to a situation where somebody who is not a doctor, but claims to have some knowledge of patient care, ends up prescribing medication."* (**Participant #10 – Senior Officer**)

*"So, these are some of the issues that are beyond our own level of you know interventions. We only advocate for more to be employed, but then people get employed after one year after a few months they lived their system and travel out for better paid job. So these are some of the challenge."* (**Participant #2 – Senior Officer**)

*"Most times, people that are in the establishment might actually have to function in a dual role. You understand? It might actually be that it's the nurse that is doubling as the pharmacist."*

(**Participant #6 – Executive**)

Six participants agreed that coordination gaps between national and state actors hinder effective AMR policy implementation. One participant emphasized that a lack of political will and leadership commitment impedes the inclusion of the private sector. Overall, government presence was described as limited across the board.

*"Yes, just to say that in all of these, it has to do with the political will. Leadership is key. This is quite important, but it's something that many developing countries don't really pay attention to."* (**Participant #13 – Senior Officer**)

Nine participants agreed that unregulated access to antibiotics enables widespread misuse. One participant highlighted the circulation of counterfeit and low-quality drugs by unqualified individuals. Others noted that some private facilities may prioritize profit over proper prescription practices.

*"In fact, you even have other categories of health workers who open private facilities. And so, antimicrobial stewardship is not their priority. Their priority is profit. As a result, if patients come to them, and they prescribe antibiotics, but the patient cannot afford the full cost of the full dose, they provide the patient only what he can afford."* (**Participant #4 – Professional Association Representative)**

*"Over-the-counter prescriptions, without a doctor's prescription, you can actually get it…So it will lead to people having access. There is free access to all these, so that could actually be part of the reason. It's part of the reason why we have this high resistance to antimicrobials."*

(**Participant #6 – Executive**)

Five participants indicated that private sector data is largely missing from national AMR tracking efforts, and one noted that while monitoring systems exist, they are underutilized. Surveillance gaps were reported to limit the ability to assess AMR trends and evaluate the impact of stewardship interventions. Half of the interviewees agreed that AMR regulations exist but are poorly enforced and minimal in impact. Over-the-counter access to antibiotics was reported as widespread and largely unregulated. One participant added that the lack of oversight enables non-compliant prescribing practices to continue unchecked.

> *"… in terms of monitoring, […] there are government agencies responsible for that, but the reality is that monitoring is not happening as well as it should. However, there are, I'll call them indicators for monitoring, but it's not happening as effectively as it should."* (**Participant #14 – Executive**)

> *"But they are not enforced so strictly... That's why they know there is no such enforcement of these laws and legislations."* (**Participant #5 – Senior Officer**)

**Opportunities.** Despite systemic limitations, participants identified growing national and global commitment to AMR, with progress in surveillance, digital information sharing, and efforts to expand stewardship training. One participant highlighted vaccination as a national strategy to reduce antibiotic demand with another noted how the Ministry of Health is actively advocating for structured, prescription-based antibiotic use to promote rational prescribing practices.

> *"As you know, we can communicate with the government and then come up with other strategies related to, you know, so that there's a structured way, so that there are algorithms provided for prescribing in the community."* (**Participant #8 – Senior Officer**)

> *"With all this and effective coordination, we can mitigate this issue of antimicrobial resistance. There's a particular body in the country—we call it the National Council on Health. This is a body that is the highest decision-making body in Nigeria, and it informs all—that is, the state of the country, including the honorable Minister. Any decision at that particular level, once it is agreed, is binding on everybody."* (**Participant #3 – Executive**)

Participants shared that Nigeria has made progress in AMR surveillance since 2017, though challenges persist in quality coordination. For example, one participant highlighted the use of digital and facility-based platforms, such as AMR Net, to support data sharing across institutions. A few other participants described the use of point prevalence surveys (PPS) in hospitals to monitor antibiotic use and inform stewardship efforts.

> *"For Nigeria, we have been doing AMR surveillance since 2017, even though it is not very optimal, and our data is not as representative as we would expect. But then we have started something."* (**Participant #2 – Senior Officer**)

> *"So, there are quite a few hospitals now beginning to implement point prevalence surveys in their centers to understand how antibiotics are being used. That is coming up, but there's still a lot that needs to be done in terms of monitoring."* (**Participant #14 – Executive**)

One participant shared that Nigeria's AMR National Action Plan, co-developed with global partners, guides stewardship and surveillance efforts and involves the private sector. Most participants pointed out that global initiatives are integrated into national policies and training programs, providing both funding and technical support.

> *"Yeah, our national team actually plans to engage all levels of care are both private and public. So the clinics you're talking about are being captured on the action plan."* (**Participant #2 – Senior Officer**)

*"You know the methodology of what you're gonna be doing. I'm sure private sectors will be part of the people that you are engaging. So those are opportunities for them to be included in the stewardship related to what we are doing."* (**Participant #6 – Executive**)

**Strategic actions.** Participants proposed actionable strategies including performance incentives, stronger regulation, greater government investment, and enhanced surveillance integration. Public-private partnerships were emphasized as a key driver of sustainable stewardship.

In particular, most participants agreed that incentives, such as awards and certifications, were seen as effective in promoting AMR leadership within private health institutions. One participant highlighted that government-supported recognition and public accountability mechanisms, including reward-and-sanction systems, can motivate individuals to adhere to AMR policies. These strategies were also noted as potential tools to support practitioner retention and reduce reliance on unqualified providers

*"This puts accountability on them. They know, "We have to do this; we are accountable." There should be a reward-and-sanction system in place because, really, if we don't have this, everything will just be business as usual. But when we want to address antimicrobial resistance, it should no longer be business as usual."* (**Participant #6 – Executive**)

*"There could be incentives offered to private facilities to engage in proper practices with respect to antibiotic use and evaluating patients properly, especially those with suspected bacterial infections. One of those incentives could be offering endorsements, or certifications, really."* (**Participant #14 – Executive**)

Participants emphasized that government involvement is essential to expanding access to affordable treatment. A few participants added that government-led certifications and regulations help to build public trust and promote system-wide adherence to AMR strategies.

*"Once we put the political willingness to implement a few of the things that I have mentioned, you understand, we have registries—doctor and nurses registries—of people that are in the public space and also the private space."* (**Participant #6 – Executive**)

One participant emphasized that effective stewardship efforts require local evidence and real-time data to guide treatment decisions. Participants suggested that expanding surveillance to include private healthcare facilities is key to ensuring comprehensive AMR tracking. Another participant highlighted the importance of standardizing prescription documentation to eliminate casual, verbal prescriptions and support traceability. They added that mandating monthly reporting could strengthen this oversight. Three participants encouraged consistent communication to promote data sharing and compliance within private facilities.

Participants agreed that strong penalties are necessary to discourage non-compliance with AMR policies. One participant emphasized the importance of involving ministries, councils, and local governments in monitoring and law enforcement to ensure regulatory success. Participants agreed that without proper enforcement, inappropriate prescribing and the circulation of low-quality drugs will continue to fuel AMR.

*"… we need the government to make a few examples of these people because doctors don't have prosecuting capacity. We are not law enforcement agents. We are simply there to guide."* (**Participant #9 – Professional Association Representative**)

*"So, if anybody is doing the wrong thing, the person can lose his or her license to practice. Because of that, that has created a kind of fear among people to do the right thing. I'm talking about the physicians, doctors, dental surgeons, and the rest."* (**Participant #12 – Professional Association Representative**)

Lastly, all participants emphasized the importance of building partnerships with regulatory bodies across government sectors. Some participants specifically highlighted the need to include the private sector in national coordination efforts and technical working groups.

*"And so, it should start with, as I mentioned, involving them in the technical working group so that, we can bring their concerns and challenges and then support them in that direction."* (**Participant #5 – Senior Officer**)

*"One of the key things that will help is not just for a particular department to act but to bring as many people on board— looking at academia, the Ministry of Agriculture, the Ministry of Health, the National Centre for Disease Control, looking at that and also our own aspect."* (**Participant #13 – Senior Officer**)

## Discussion

This study explored barriers, facilitators and opportunities for designing and implementing antibiotic stewardship interventions in Nigeria's private healthcare sector. Interviews with national-level interest holders identified four actor groups - private sector, practitioners, government/health systems, and patients – each playing a central role in stewardship efforts. Participants described inconsistent prescribing, gaps in regulation and training, limited government cooperation, and low public awareness as key contributors to AMR. They also identified actionable opportunities for intervention, including improved training programs, broader and more effective use of guidelines and standard operating procedures, public-partner partnerships, and expanded awareness campaigns.

Our findings are consistent with previous studies [17], which confirms that most Nigerians rely on private healthcare services and often pay out of pocket. Despite this, the private sector has remained largely excluded from formal health planning and has not been prioritized in past stewardship programs [18]. Of note, participants emphasized the openness and willingness of private providers to adopt new practices.

Many participants in our study emphasized the importance of engaging the private sector early in the planning of national stewardship efforts. This aligns with existing research showing that effective public-private collaboration is critical to scaling stewardship activities in LMICs [19]. Participants also highlighted the role of regulatory bodies, consistent with literature indicating that regulatory involvement enhances program sustainability [, 20–22]. For example, a 2021 case study from India demonstrated the feasibility of co-developing clinical guidelines and training through partnerships between government and over 20 private institutions [23].

At the practitioner level, participants in our study repeatedly mentioned inconsistent prescribing habits among private practitioners, including precautionary overuse and prescribing without clear clinical indication, which are patterns also documented in prior studies [10]. A 2023 survey of 252 Nigerian doctors including public and private practitioners found that only 41% had good AMR knowledge, 16% had positive attitudes, and just 6% exhibited appropriate prescribing practices [24]. These results underscore the need for enhanced training, stronger stewardship programs, and institutional support to improve antibiotic use and reduce resistance [24].

Training gaps were a recurring theme. Participants noted that many prescribers lack foundational AMR knowledge and that training opportunities remain limited, particularly in the private sector. These observations are consistent with a 2022 scoping review of national stewardship activities across eight African countries, including Nigeria, which identified workforce training as a persistent challenge [25]. Evidence from other LMICs suggests that well-designed training, when paired with institutional support, can improve prescribing and reduce resistance [26–27]. Our participants further emphasized

the importance of diagnostic testing (e.g., culture) prior to initiating antibiotics, echoing findings from South Africa where limited access to diagnostics was found to drive empirical prescribing [28].

In addition to training gaps, participants reported critical human resource shortages, which stem from insufficient pay and inadequate leave benefits primarily affecting Nigeria's private healthcare sector. Dual-role staffing and uneven workforce distribution across facilities were also noted as challenges. These findings are consistent with research by Yakubu et al. [29], which documented widespread health worker migration from Nigeria due to poor remuneration, working conditions, and systemic underinvestment in the health system. Our study participants suggested that incentive programs (e.g., awards and certifications) could help promote stewardship leadership in private institutions. This is supported by a 2020 study from Malawi, which found that identifying stewardship champions and offering institutional recognition were key facilitators for successful implementation of the WHO's stewardship toolkit in low-resource settings [30].

Government support was consistently identified as essential. In line with existing literature [17,20,31,32], participants stressed the need for political will, adequate funding, and stronger regulatory enforcement to address barriers such as limited antibiotic availability, poor laboratory infrastructure, and workforce shortages. Concerns about unregulated antibiotic access were also widespread. Participants acknowledged that, while antibiotic prescribing regulations exist, enforcement is weak and has minimal impact. Participants advocated for more robust enforcement mechanisms and stricter penalties to deter non-compliance. Given that weak governance and inadequate regulation are recognized drivers of AMR, strong political commitment and legislative action remain critical [23].

Surveillance emerged as another significant gap, particularly the lack of data from the private sector. Participants noted that Nigeria's current surveillance infrastructure is skewed toward tertiary and government laboratories, limiting representativeness, a concern echoed in previous research [33]. Collecting high-quality, representative data is essential for an effective AMR response [34–35]. A 2020 study by Médecins Sans Frontières demonstrated that decentralized AMR surveillance using a portable Mini-Lab in a low-resource hospital in Haiti was feasible and generated valuable data to inform national response efforts [36].

Participants consistently described how financial hardship leads many patients to self-medicate, purchase partial doses, or rely on unregulated vendors—issues common in many low-resource settings [37]. A 2019 national survey found that 31% of Nigerians took antibiotics without a prescription, only 8.3% had good AMR knowledge, and over 76% felt powerless to prevent its spread [38]. These findings underscore the urgent need for targeted public education and awareness initiatives.

Insights from our in-depth interviews highlight that sustainable behavioral change requires interventions targeting multiple levels simultaneously. Our findings mirror recent Nigerian studies documenting minimal stewardship activities in private healthcare facilities and substantial knowledge gaps among healthcare providers [10,39]. Applying the Social-Ecological Model enabled us to identify intervention points across individual, organizational, community, and policy levels – critical for addressing AMR in Nigeria's fragmented healthcare system, where structural barriers compound individual behavior challenges.

This study has several strengths. By sampling individuals from prominent national organizations in Nigeria, we captured valuable insights into the under-researched role of the country's private sector in AMR response efforts. Using the Braun and Clarke's thematic analysis framework [15], we examined the participants' perspectives pertaining to system, practitioner, and patient-level themes. The study followed a pre-specified protocol, adhering to methodological standards for conducting qualitative research in global health contexts and ensuring rigorous documentation for enhanced reproducibility. However, limitations include a small sample size and the predominance of senior-level participants, which may have excluded frontline perspectives. While patient behavior was a major theme, no patients were interviewed. As with most qualitative studies, the reliance on self-reported data may introduce potential recall and reporting bias. Additionally, given the seniority and professional affiliations of participants, responses may have been influenced by social desirability bias, with individuals potentially presenting their organizations or practices in a more favorable light. To mitigate this, interviews

were conducted in private settings and framed to encourage candid reflection. Furthermore, the interviewers were affiliated with Canadian institutions and were not Nigerian, which may have influenced participant responses. To address this, interviews were organized in close collaboration with Nigerian research partners who facilitated recruitment and scheduling, and all interviewers received training in global health research best practices, including culturally sensitive interviewing and reflexivity.

This study contributes novel insights into the feasibility of stewardship implementation in Nigeria's private healthcare sector by centering the perspectives of national-level interest holders – a group rarely represented in AMR research. Their reflections revealed both systemic constraints and institutional readiness, offering a foundation for future intervention design. Building on these findings, our next step involves developing and piloting a tailored educational intervention for private sector physicians, with the goal of evaluating its acceptability and impact. Future research should also incorporate frontline providers and patients to capture operational realities and behavioral drivers more fully. Expanding the evidence base in this way will be critical to developing inclusive, scalable stewardship strategies that reflect the complexity of Nigeria's mixed health system.

## Conclusions

This study highlights critical factors influencing AMR response and stewardship planning in Nigeria's private healthcare sector. Participants emphasized the central role of private providers in healthcare delivery and their current underrepresentation in national AMR strategies. Key barriers identified included inconsistent prescribing practices, limited diagnostic capacity, workforce shortages, weak regulatory enforcement, and low public awareness. Despite these challenges, participants expressed strong interest in stewardship initiatives and identified actionable opportunities for intervention. These include standardized training programs, public-private collaboration, stronger regulatory framework, performance-based incentives, and community-level awareness campaigns. Ultimately, our findings underscore the urgent need for a context-specific, sustainable stewardship program for the private sector in order to strengthen AMR control and ensure equitable, effective antibiotic use across all levels of the health system.

## Supporting information

**S1 File. COREQ checklist.**
(PDF)

**S2 File. Interview guide.**
(PDF)

**S3 File. Themes and Subthemes.**
(PDF)

## Acknowledgments

The authors would like to sincerely thank all study participants for generously sharing their time, insights, and expertise.

## Author contributions

**Conceptualization:** Nnakelu Eriobu, Blessing Uche, Charity Oga-Omenka, Giorgia Sulis.

**Data curation:** Aaya Mahdi.

**Formal analysis:** Aaya Mahdi, Sarah Pascale Ngassa Detchaptche, Joel Shyam Klinton.

**Funding acquisition:** Giorgia Sulis.

**Methodology:** Aaya Mahdi, Sarah Pascale Ngassa Detchaptche, Joel Shyam Klinton, Charity Oga-Omenka, Giorgia Sulis.

**Project administration:** Nnakelu Eriobu, Blessing Uche, Nawal Maredia, Giorgia Sulis.

**Resources:** Giorgia Sulis.

**Supervision:** Giorgia Sulis.

**Visualization:** Aaya Mahdi.

**Writing – original draft:** Aaya Mahdi.

**Writing – review & editing:** Aaya Mahdi, Sarah Pascale Ngassa Detchaptche, Joel Shyam Klinton, Nnakelu Eriobu, Blessing Uche, Nawal Maredia, Charity Oga-Omenka, Giorgia Sulis.

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
