## [Decision Letter · Decision Letter 0]

29 Sep 2025

PGPH-D-25-02175

Barriers and facilitators to antibiotic stewardship in Nigeria’s private healthcare sector: a qualitative interview study with key interest holders

Dear Dr.Giorgia Sulis,

Thank you for submitting your manuscript to PLOS Global Public Health. After careful consideration, we feel that it has merit but does not fully meet PLOS Global Public Health’s publication criteria as it currently stands. Therefore, we invite you to submit a revised version of the manuscript that addresses the points raised during the review process.

We look forward to receiving your revised manuscript.

Kind regards,

Muhammad Asaduzzaman, MD MPH MPhil

Academic Editor

Journal Requirements:

1. Please ensure that the Title in your manuscript file and the Title provided in your online submission form are the same.

2. Please send a completed 'Competing Interests' statement, including any COIs declared by your co-authors. If you have no competing interests to declare, please state "The authors have declared that no competing interests exist". Otherwise please declare all competing interests beginning with the statement "I have read the journal's policy and the authors of this manuscript have the following competing interests:"

3. Please provide separate figure files in .tif or .eps format.

Additional Editor Comments (if provided):

Reviewers' comments:

Reviewer's Responses to Questions

**Comments to the Author**

1. Does this manuscript meet PLOS Global Public Health’s publication criteria?

Reviewer #1: Yes

Reviewer #2: Yes

2. Has the statistical analysis been performed appropriately and rigorously?

Reviewer #1: N/A

Reviewer #2: N/A

3. Have the authors made all data underlying the findings in their manuscript fully available (please refer to the Data Availability Statement at the start of the manuscript PDF file)?

Reviewer #1: No

Reviewer #2: Yes

4. Is the manuscript presented in an intelligible fashion and written in standard English?

Reviewer #1: Yes

Reviewer #2: Yes

Reviewer #1: Thank you for the opportunity to review this manuscript on antimicrobial resistance (AMR) and stewardship efforts in Nigeria’s private healthcare sector. This is an important and timely contribution, as AMR poses a major global health threat, with particularly severe consequences in low- and middle-income countries where surveillance, regulation, and access to diagnostics remain limited. Your manuscript addresses an important and under-researched issue: The study is well-motivated, methodologically sound, and written with clarity. The integration of perspectives from senior national and regional stakeholders is a strength, and the use of the Social-Ecological Model provides a useful framework to interpret findings. The article also situates results within the wider LMIC context, making it relevant to a global readership. Following are areas that require improvement before the paper is ready for publication:

1 Title

Please specify which groups were interviewed and consistently refer to them as stakeholders to make the title clearer and more precise.

2. Abstract

-- Provide a rationale for why participants were purposively selected.

- Improve how results are reported and explicitly link them to the SEM framework.

- Clearly distinguish between barriers and facilitators, or present broader, overarching themes.

- Refine the conclusion to avoid repetition and provide a concise, actionable call to action.

3 Introduction

- Avoid repeatedly citing statistics (e.g., AMR burden globally and in LMICs, or private sector underrepresentation). Streamline to maintain focus and readability.

- Use one consistent term, either interest holders or stakeholders.

- Improve readability by using shorter paragraphs and reducing the number of references per paragraph.

3. Methods

- Explain why 15 participants were considered sufficient (e.g., data saturation, representation).

- Table 1 is difficult to interpret; consider moving it to the appendix or summarizing participants by category and sample size.

- Clarify whether interviews were conducted in English only or if translation was required.

- Describe any pilot testing or refinement of the interview guide during data collection.

- Expand on researcher positionality and reflexivity: Were interviewers Nigerian? How might institutional affiliations (Ottawa vs. Nigeria) have influenced responses?

- Provide the dates and approval numbers for ethical review.

4. Data Analysis

- Clarify whether the coding-to-theme structure was developed inductively from the data or applied deductively.

- The sequence from codes to subthemes to SEM is currently confusing; consider simplifying for clarity.

- Explain the combined use of NVivo and Excel, e.g., NVivo for coding and Excel for organizing/exporting results, to avoid perceived duplication.

5. Results

- Review the results for clarity, conciseness, and alignment with the study objectives.

- Ensure themes are clearly linked to the research questions and SEM framework.

- Consider moving detailed tables to supporting materials and providing a summarized description of themes and - subthemes in the main text.

- Revise subtheme associations to enhance comprehensibility.

6. Discussion

- The discussion is overly long and reference-heavy. Streamline by emphasizing what the study uniquely adds rather than summarizing multiple external studies.

- Highlight new insights that were not previously documented and explicitly discuss how findings advance or challenge existing knowledge.

- Strengthen policy and practice implications: specify what Nigerian policymakers and private providers should prioritize and what is feasible.

- Acknowledge potential social desirability bias due to senior-level participants and clarify how the absence of frontline providers and patients limits interpretation of private-sector dynamics.

Reviewer #2: The paper reads well. However, the analysis is very surface level in the sense that the results read like a list. For example, the paper listed barriers but have not gone deeper into the analysis, looking at the differences between the narratives of the participants in terms of their work experience and responsibilities. For examples, the ones from the private sector, did they give different response in some areas compared to other participants? Also, the way barriers have been clustered together for providers and patients, it is not clear which are the most significant barriers, for example. The authors only present the barrier and the number of people reporting that. The discussion also does not talk about implications of the findings. The authors mostly compare their findings with other studies.

**Do you want your identity to be public for this peer review?** For information about this choice, including consent withdrawal, please see our Privacy Policy

Reviewer #1: No

Reviewer #2: No

---

## [Decision Letter · Decision Letter 1]

18 Nov 2025

PGPH-D-25-02175R1

Barriers and facilitators to antibiotic stewardship in Nigeria’s private healthcare sector: a qualitative interview study with national health and regulatory interest holders

Dear Dr. Giorgia Sulis,

Thank you for submitting your manuscript to PLOS Global Public Health. After careful consideration, we feel that it has merit but does not fully meet PLOS Global Public Health’s publication criteria as it currently stands. Therefore, we invite you to submit a revised version of the manuscript that addresses the points raised during the review process.

We look forward to receiving your revised manuscript.

Kind regards,

Muhammad Asaduzzaman, MD MPH MPhil

Academic Editor

Journal Requirements:

Reviewers' comments:

Reviewer's Responses to Questions

**Comments to the Author**

Reviewer #1: All comments have been addressed

Reviewer #2: All comments have been addressed

publication criteria?

Reviewer #1: Yes

Reviewer #2: Yes

3. Has the statistical analysis been performed appropriately and rigorously?

Reviewer #1: N/A

Reviewer #2: N/A

4. Have the authors made all data underlying the findings in their manuscript fully available (please refer to the Data Availability Statement at the start of the manuscript PDF file)?

Reviewer #1: Yes

Reviewer #2: Yes

5. Is the manuscript presented in an intelligible fashion and written in standard English?

Reviewer #1: Yes

Reviewer #2: Yes

Reviewer #1: All the comments have been well addressed.

Reviewer #2: **The authors addressed my comment well. One minor revision needs to be made: Remove the quotation in L706 from the discussion section. Not needed. We simply discuss in discussion section, not introduce any new result. It is okay to put quant data but we usually do not put qualitative data (quotes) in discussion.**

**Do you want your identity to be public for this peer review?** For information about this choice, including consent withdrawal, please see our Privacy Policy

Reviewer #1: No

Reviewer #2: No

---

## [Editor Report · Decision Letter 2]

11 Dec 2025

Barriers and facilitators to antibiotic stewardship in Nigeria’s private healthcare sector: a qualitative interview study with national health and regulatory interest holders

PGPH-D-25-02175R2

Dear Giorgia Sulis,

We are pleased to inform you that your manuscript 'Barriers and facilitators to antibiotic stewardship in Nigeria’s private healthcare sector: a qualitative interview study with national health and regulatory interest holders' has been provisionally accepted for publication in PLOS Global Public Health.

Best regards,

Muhammad Asaduzzaman, MD MPH MPhil

Academic Editor